# Spontaneous neurotransmission signals through store-driven Ca$^{2+}$ transients to maintain synaptic homeostasis

Austin L Reese[1], Ege T Kavalali[1,2]*

[1]Department of Neuroscience, University of Texas Southwestern Medical Center, Dallas, United States; [2]Department of Physiology, University of Texas Southwestern Medical Center, Dallas, United States

**Abstract** Spontaneous glutamate release-driven NMDA receptor activity exerts a strong influence on synaptic homeostasis. However, the properties of Ca$^{2+}$ signals that mediate this effect remain unclear. Here, using hippocampal neurons labeled with the fluorescent Ca$^{2+}$ probes Fluo-4 or GCAMP5, we visualized action potential-independent Ca$^{2+}$ transients in dendritic regions adjacent to fluorescently labeled presynaptic boutons in physiological levels of extracellular Mg$^{2+}$. These Ca$^{2+}$ transients required NMDA receptor activity, and their propensity correlated with acute or genetically induced changes in spontaneous neurotransmitter release. In contrast, they were insensitive to blockers of AMPA receptors, L-type voltage-gated Ca$^{2+}$ channels, or group I mGluRs. However, inhibition of Ca$^{2+}$-induced Ca$^{2+}$ release suppressed these transients and elicited synaptic scaling, a process which required protein translation and eukaryotic elongation factor-2 kinase activity. These results support a critical role for Ca$^{2+}$-induced Ca$^{2+}$ release in amplifying NMDA receptor-driven Ca$^{2+}$ signals at rest for the maintenance of synaptic homeostasis.

*For correspondence: ege.kavalali@utsouthwestern.edu

**Competing interests:** The authors declare that no competing interests exist.

## Introduction

Studies in the last decade have shown that spontaneous release events trigger biochemical signaling leading to maturation and stability of synaptic networks, local dendritic protein synthesis and control postsynaptic responsiveness during homeostatic synaptic plasticity (*Chung and Kavalali, 2006*; *Sutton et al., 2006*; *Kavalali, 2015*). Most surprisingly, these studies have demonstrated that postsynaptic excitatory receptor blockade or inhibition of neurotransmitter release in addition to action potential blockade induces faster and more pronounced homeostatic synaptic potentiation (*Sutton et al., 2006*; *Nosyreva et al., 2013*). There is evidence that alterations in resting Ca$^{2+}$ signaling, partly triggered via activation of NMDA receptors at rest, is critical for these effects (*Wang et al., 2011*; *Nosyreva et al., 2013*). However, to date there is no direct information on the properties of dendritic Ca$^{2+}$ signals elicited by spontaneous release events under physiological circumstances. Our group's previous work as well as others used electrophysiology to show that, indeed, under physiological levels of extracellular Mg$^{2+}$ spontaneous miniature excitatory post-synaptic currents (mEPSCs) possess a sizable NMDA receptor-mediated component, indicating that NMDA receptors signal at rest under physiological conditions without requiring local AMPAR-mediated dendritic depolarizations (*Espinosa and Kavalali, 2009*; *Povysheva and Johnson, 2012*; *Gideons et al., 2014*). The existence of an NMDA component within mEPSCs agrees with earlier estimates of incomplete Mg$^{2+}$ block of canonical NMDA receptors near resting membrane potentials, and therefore it does not necessarily involve NMDA receptor subunits with altered Mg$^{2+}$ sensitivity (*Jahr and Stevens, 1990*). Nevertheless, the NMDA receptor Ca$^{2+}$ influx under these conditions is estimated to be small, corresponding to approximately 20% of the full Ca$^{2+}$ influx carried by

**eLife digest** Learning and memory is thought to rely on changes in the strength of the connections between nerve cells. When an electrical impulse travelling through a nerve cell reaches one of these connections (called a synapse), it causes the cell to release chemical transmitter molecules. These bind to receptors on the cell on the other side of the synapse. This starts a series of events that ultimately leads to new receptors being inserted into the membrane of this second cell, which strengthens the connection between the two cells.

The receptors involved in this process belong to two groups, called AMPA and NMDA receptors. Both groups are ion channels that regulate the flow of charged particles from one side of a cell's membrane to the other. In resting nerve cells, NMDA receptors are partially blocked by magnesium ions. However, the binding of the transmitter molecules to AMPA receptors causes these receptors to open and allow positively charged sodium ions into the cell. This changes the electrical charge across the cell membrane, which displaces the magnesium ions from the NMDA receptors so that they too open. Calcium ions then enter the cell through the NMDA receptors and activate a signaling cascade that leads to the production of new AMPA receptors.

Nerve cells also release transmitter molecules in the absence of electrical impulses, and evidence suggests that individual cells can use this 'spontaneous transmitter release' to adjust the strength of their synapses. When these spontaneous release levels are high, AMPA receptors are removed from the membrane of the nerve after the synapse to make it less sensitive to the transmitter molecules. Conversely, when spontaneous release levels are low, additional AMPA receptors are added to the membrane to increase the sensitivity.

Reese and Kavalali have now identified the mechanism behind this process by showing that spontaneously released transmitter molecules cause small amounts of calcium to enter the second nerve cell through NMDA receptors, even when these receptors are blocked by magnesium ions. This trickle of calcium triggers the release of more calcium from stores inside the cell, which amplifies the signal. The ultimate effect of the flow of calcium into the cell is to block the production of AMPA receptors, and ensure that the synapse does not become any stronger. As confirmation of this mechanism, Reese and Kavalali showed that simulating low levels of spontaneous activity by blocking the so-called 'calcium-induced calcium release' has the opposite effect. This led to more AMPA receptors being produced and stronger synapses. Taken together these findings indicate that spontaneous transmitter release exerts an outsized influence on communication between neurons by maintaining adequate levels of AMPA receptors via these 'amplified' calcium signals.

unblocked NMDA receptors (*Espinosa and Kavalali, 2009*). Therefore, as NMDA receptor-driven $Ca^{2+}$ signals at rest are expected to be relatively minor in magnitude, it remains unclear how their blockade could be critical in producing homeostatic synaptic scaling. To address this question, we visualized the resting NMDA receptor-driven $Ca^{2+}$ signals and found that they are amplified by a $Ca^{2+}$-induced $Ca^{2+}$ release mechanism to elicit downstream signaling events. Importantly, based on this information, we also show that direct suppression of these resting $Ca^{2+}$ signals is sufficient to elicit eEF2 kinase dependent postsynaptic scaling.

## Results

### Visualization of miniature spontaneous $Ca^{2+}$ transients in hippocampal neurons

To detect transient $Ca^{2+}$ signals that occur under resting conditions—in the absence of action potentials—we took advantage of the $Ca^{2+}$ indicator dye Fluo-4 or the $Ca^{2+}$ sensitive fluorescent protein GCaMP5K as reporters (*Gee et al., 2000*; *Akerboom et al., 2012*). To visualize synapses, both reporters were used on hippocampal neurons that were infected with lentivirus expressing the fusion protein Synaptobrevin2-mOrange (Syb2-mOrange) consisting of a chimera of the synaptic vesicle protein synaptobrevin2 with the pH sensitive red-shifted fluorophore mOrange (*Ramirez et al., 2012*) (*Figure 1A–D*). In these experiments, the signal contribution of Syb2-mOrange during live imaging is negligible (see *Figure 2—figure supplement 1*). In Fluo-4 experiments, neurons were

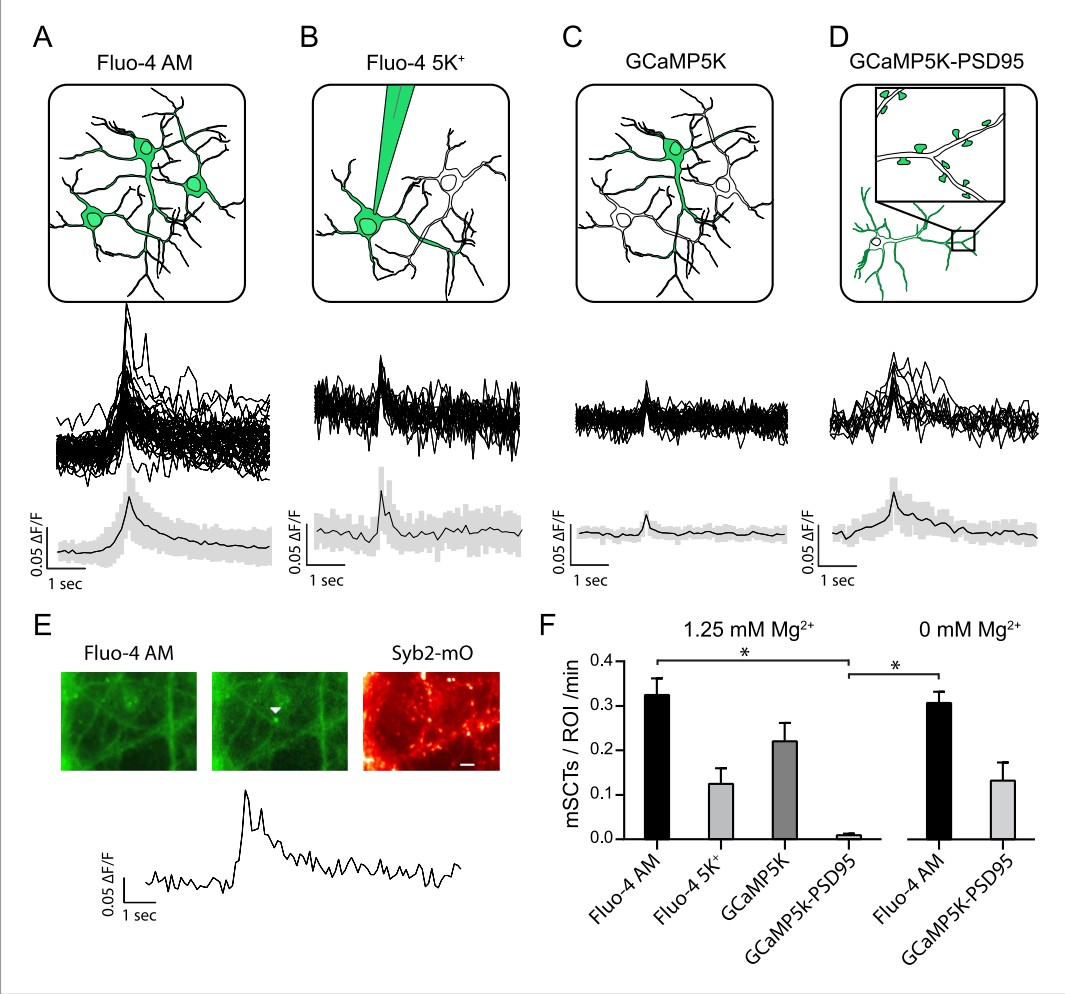

**Figure 1**. Multiple approaches to detect miniature spontaneous $Ca^{2+}$ transients (mSCTs) in the presence of TTX and physiological $Mg^{2+}$. (**A**) Loading dissociated rat hippocampal cultures with Fluo-4 AM dye labels all cells on the coverslip (above) and produces the largest signal amplitudes, shown as example traces and an average with standard deviation (below) N = 38 experiments, 7 cultures. (**B**) Individual neurons were loaded with the salt form of Fluo-4 at the whole cell recording configuration via a pipette containing 200 µM of the dye. N = 4 experiments, 1 culture. (**C**) Low efficiency lipotransfection with the highly sensitive GCaMP5K variant produces sparse labeling of neurons across the coverslip but low signal ($\Delta F/F$) amplitudes. N = 5 experiments, 1 culture. (**D**) Lentiviral mediated transfection with GCaMP5K-PSD95 targets the fluorescent construct to the postsynaptic densities of all cells on each coverslip. N = 15 experiments, 4 cultures. (**E**) Example images and trace of a mSCT visualized with Fuo-4 AM and its corresponding Syb2-mOrange puncta. Panels show baseline and peak fluorescence intensity with the arrow marking peak fluorescence intensity of the mSCT. Scale bar 5 µm. (**F**) Frequencies expressed as mSCTs per ROI per minute show the highest efficiency of mSCT detection with Fluo-4 AM. Fluo-4 AM based experiments performed with no $Mg^{2+}$ in the external solution reported no significant changes in mSCT compared to the presence $Mg^{2+}$. The postsynaptically localized reporter GCaMP5K-PSD95 reports statistically lower frequencies when compared to Fluo-4 AM (Fluo-4 AM 1.25 mM $Mg^{2+}$ vs GCaMP5K-PSD95 1.25 mM $Mg^{2+}$ p = 0.0003, Fluo-4 AM 0 mM $Mg^{2+}$ vs GCaMP5K-PSD95 1.25 mM $Mg^{2+}$ p = 0.0031, via one-way ANOVA with Holm-Sidak's multiple comparisons) 0 mM $Mg^{2+}$ Fluo-4 AM N = 16 experiments, 8 cultures. 0 mM $Mg^{2+}$ GCaMP5kK-PSD95 N = 10 experiments, 4 cultures.

initially incubated and labeled with the membrane permeable analog of Fluo-4 (Fluo-4 AM) (*Figure 1A*) followed by dye removal and perfusion with a Tyrode's solution containing 2 mM $Ca^{2+}$, 1.25 mM $Mg^{2+}$ as well as 1 µM tetrodotoxin (TTX) to block action potentials. Fluorescence images were collected at a frequency of 10 Hz and fluorescence intensity traces were generated for the regions of interest (ROIs) selected over Syb2-mOrange puncta which fluorescence was maximized at

the end of each experiment using 50 mM $NH_4Cl$ (*Figure 1E*). Under these conditions, we could detect rapid $Ca^{2+}$ transients (miniature spontaneous calcium transients or mSCTs) with absolute values that were at least 2 standard deviations above the mean of the preceding baseline period (2 s) (*Figure 1F*). These events occurred at a frequency of $0.32 \pm 0.04$ $min^{-1}$ per ROI, consistent with earlier estimates of the frequency of spontaneous fusion events per release site (*Leitz and Kavalali, 2014*). Repeating the same experimental protocol with Fluo-4 AM in the absence of $Mg^{2+}$ did not yield a significantly different mSCT frequency (*Figure 1F*) suggesting that under physiological $Mg^{2+}$ concentrations we could detect a majority of mSCTs. Interestingly, even though the presence of extracellular $Mg^{2+}$ is expected to greatly diminish NMDAR current magnitudes (*Espinosa and Kavalali, 2009*; *Gideons et al., 2014*), imaging experiments did not reveal a significant difference in mSCT amplitudes detected in $Mg^{2+}$ (1.25 mM $Mg^{2+}$ $\Delta F/F_o = 0.063 \pm 0.001$, 0 mM $Mg^{2+}$ $\Delta F/F_o = 0.067 \pm 0.002$, p = 0.16, Student's unpaired t-test, N = 825 events from 8 experiments). The fact that mSCT amplitude was unaffected by extracellular $Mg^{2+}$ indicates mSCTs measured by Fluo-4 AM were not likely to be solely dependent on NMDA receptor activity.

In parallel experiments, we delivered the salt form of Fluo-4 (200 µM) with a patch pipette in the whole-cell recording configuration and performed the same imaging protocol as above (*Figure 1B*). In this setting, we detected a lower frequency of events ($0.125 \pm 0.035$ $min^{-1}$ per ROI), indicating that some of the mSCTs may be susceptible to postsynaptic dialysis and wash out of soluble factors (*Figure 1F*). In agreement with this premise, when the same optical recording conditions were applied to neurons expressing a soluble version of the green emission $Ca^{2+}$ indicator probe GCaMP5K, we could detect a higher frequency of mSCTs ($0.230 \pm 0.04$ $min^{-1}$ per ROI).

In subsequent experiments, we expressed a fusion construct of GCaMP5K with the postsynaptic scaffolding protein PSD95 (GCAMP5K-PSD95) in order to target the calcium sensor specifically to the postsynaptic density (*Figure 1D*). In the presence of extracellular $Mg^{2+}$ based on the population average this setting provided the lowest estimate for the mSCT frequency ($0.009 \pm 0.004$ $min^{-1}$ per ROI) (*Figure 1F*). In contrast, removal of $Mg^{2+}$ augmented the mSCT detection rate to a level comparable to the rates we observed with Fluo-4 or soluble GCaMP5K (*Figure 1E*). This finding suggests that, in the presence of $Mg^{2+}$, postsynaptically localized GCaMP5K-PSD95 has limited ability to detect the $Ca^{2+}$ signals generated in its vicinity via $Ca^{2+}$ influx. However, experiments in the absence of $Mg^{2+}$ indicate that this probe is functional and can in principle detect these spontaneous local $Ca^{2+}$ transients as reported earlier (*Leitz and Kavalali, 2014*).

Recording in the presence of 1.25 mM $Mg^{2+}$ and 1 µM TTX, we could detect spontaneously generated $Ca^{2+}$ transients in the dendrites of hippocampal pyramidal cells with all four techniques. Although, each probe reports a different frequency these differences are statistically insignificant except when considering the difference between Fluo-4 AM and GCaMP5K-PSD95 (*Figure 1F*). Relatively lower detection efficiency of GCAMP5K-PSD95 compared to soluble probes illustrates that the majority of these transients are not localized to the postsynaptic density. The failure of $Mg^{2+}$ to decrease mSCT amplitudes as measured with Fluo-4 AM strongly suggest that a majority of transients are generated by a signaling process downstream of $Ca^{2+}$ entry rather than reporting the $Ca^{2+}$ influx per se. In order to identify the nature of this signaling, in subsequent experiments, we used the Fluo-4 AM based imaging to test conditions that alter mSCTs.

## The generation of mSCTs requires NMDA receptor mediated $Ca^{2+}$ influx

To characterize mSCTs, neurons were labeled with Fluo-4 AM as in *Figure 1A* and imaged in Tyrode's solution containing TTX (*Figure 2A*). $Ca^{2+}$ transients were detected by the slope of the rising phase as well as the peak amplitude. To ensure that these detected peaks were not noise, only mSCTs with a peak amplitude 2 standard deviations greater than the signal average of the previous 2 s were counted. *Figure 2B* shows the rise and decay times as well as the fluorescence amplitudes of 306 mSCTs identified from 6 experiments. In these experiments, the mean rise time was 0.38 s with a median of 0.29 s. The mean decay time was 0.86 s with a median of 0.47 s. The amplitude distribution had an average $\Delta F/F_o$ of 0.061 with a median of 0.049 (*Figure 2B*).

Next we tested whether NMDA receptor activity is required for the generation of mSCTs. For this purpose, synaptic ROIs were imaged in three steps. First optical recordings were obtained in Tyrode's solution with nominal $Ca^{2+}$ containing 1 µM TTX followed by the addition of 2 mM $Ca^{2+}$ and finally in Tyrode's solution containing TTX + 2 mM $Ca^{2+}$ + 50 µM AP5 (*Figure 2C*). In the absence of $Ca^{2+}$ in the

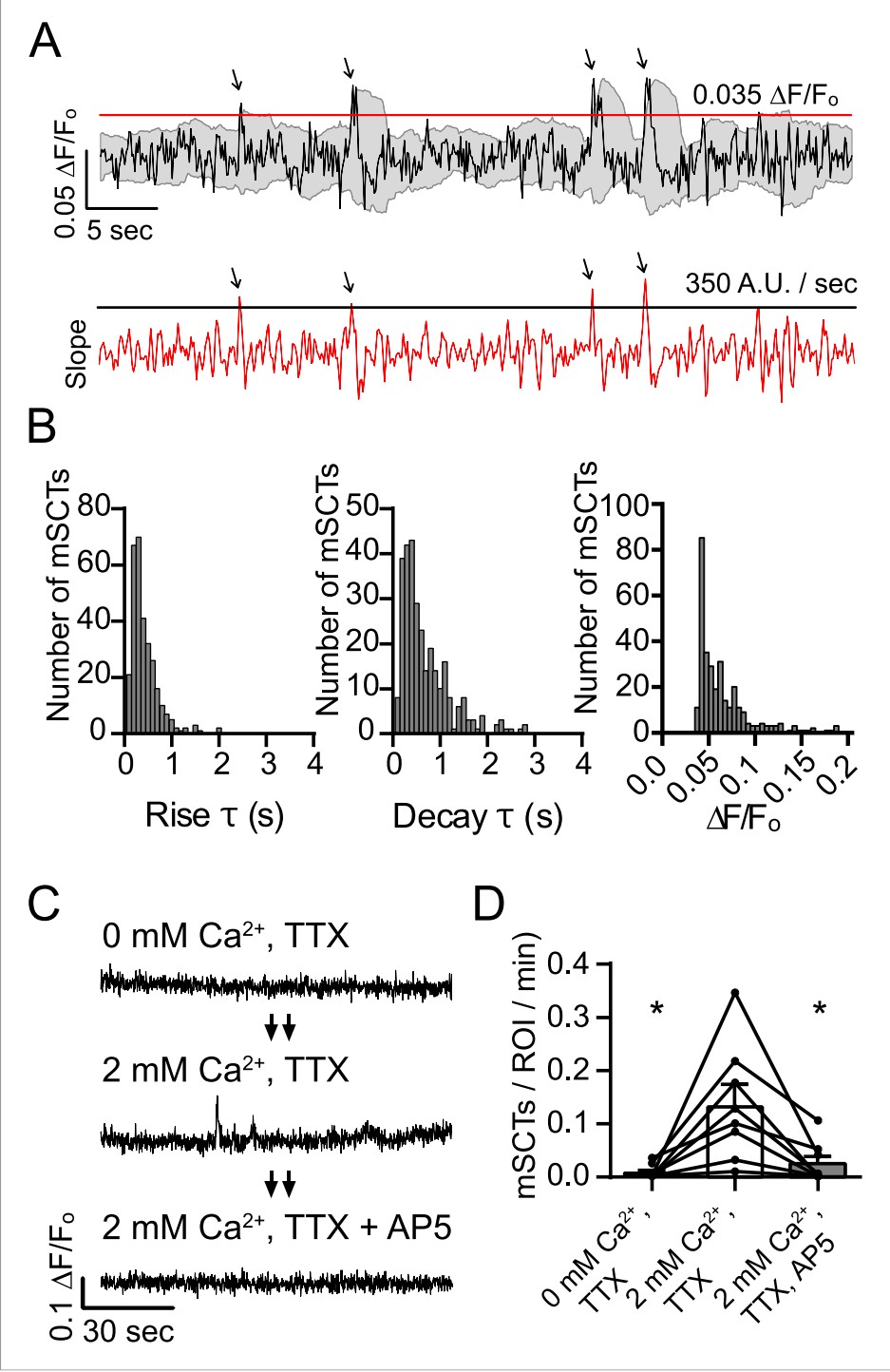

**Figure 2**. Detection and characterization of spontaneous $Ca^{2+}$ transients in physiological concentrations of $Mg^{2+}$. (**A**) Events detected from Fluo-4 AM traces having rising slope greater than 350 fluorescence units/s and a peak $\Delta F/F_o$ greater than 0.035 were counted if the peak fluorescence value was 2 standard deviations greater than the mean of the signal 2 s previous. Gray shaded region indicates the moving average plus/minus two standard deviations and the red line indicates the 0.035 $\Delta F/F_o$ threshold. Red trace shows the 2 point slope with the black line as the 350 A.U./second detection threshold. Arrows indicate peaks that satisfy these criteria. (**B**) Histograms showing rise time ($\tau$), decay time ($\tau$) and amplitudes ($\Delta F/F_o$) of mSCTs. N = 306 mSCTs from 6 experiments and 2 cultures. (**C, D**) Traces from cells (**C**) and $Ca^{2+}$ transient frequencies (**D**) were obtained by imaging first in Tyrode's solution containing no $Ca^{2+}$, then in Tyrode's containing 2 mM $Ca^{2+}$ and finally Tyrode's containing 2 mM $Ca^{2+}$ and the

*Figure 2. continued on next page*

*Figure 2. Continued*

NMDA receptor blocker AP5. Removal of extracellular $Ca^{2+}$ or block of the NMDA receptor resulted in a significant reduction in $Ca^{2+}$ transient frequency (2 mM $Ca^{2+}$ vs 0 mM $Ca^{2+}$ p = 0.038, 2 mM $Ca^{2+}$ vs 2 mM $Ca^{2+}$ + AP5 p = 0.038, via 1-way ANOVA with Holm-Sidak's multiple comparisons test). N = 8 experiments, 2 cultures.

The following figure supplement is available for figure 2:

**Figure supplement 1**. Syb2-mOrange fluorescence does not contaminate Fluo-4 AM signals.

bath, mSCTs were virtually undetectable suggesting that $Ca^{2+}$ influx is required for their generation. Switching the Tyrode's solution to TTX + 2 mM $Ca^{2+}$ brought the mSCT frequency back to normal levels, and subsequent addition of the NMDA receptor antagonist AP5 again decreased the mSCT frequency to very low levels that were not statistically different from the nominal $Ca^{2+}$ condition (*Figure 2D*). These results indicate that $Ca^{2+}$ influx through the NMDA receptor is critical for the generation of mSCTs.

## mSCTs are driven by spontaneous neurotransmitter release

To examine whether the NMDA receptor openings driving mSCTs were due to spontaneous glutamate release we took two complementary approaches. First, we took advantage of the fact that the acute application of 100 mM hypertonic sucrose is known to produce an increase in mEPSCs (*Fatt and Katz, 1952*; *Rosenmund and Stevens, 1996*). To measure this effect, hippocampal pyramidal cells were voltage clamped at −70 mV while a baseline AMPA mEPSC frequency was collected in Tyrode's solution containing 1 μM TTX, 50 μM PTX and 50 μM AP5 for 2 min. Perfusion was then switched to Tyrode's containing 100 mM hypertonic sucrose as the recording continued for 2 min. Quantification of these recordings revealed a 2.6-fold increase in mEPSC frequency upon the addition of hypertonic sucrose (*Figure 3A*). To test whether the increase in mEPSC frequency could drive an increase in mSCT frequency the experiment was repeated in neurons loaded with Fluo-4 AM. The baseline was collected in Tyrode's solution containing only TTX before changing to a solution containing TTX + 100 mM sucrose. The addition of hypertonic sucrose produced a 2.3 fold increase in mSCT frequency (*Figure 3B*), which supports the hypothesis that spontaneous glutamate release can drive the generation of postsynaptic calcium transients.

Next, to assess whether a decrease in mEPSC frequency would correlate with a decrease in mSCT frequency, we utilized neurons from mice lacking the critical SNARE-mediated fusion machinery component SNAP-25 (*Washbourne et al., 2002*). These mice die at birth; however, hippocampal neurons cultured from embryonic mice form synapses and manifest a virtual absence of evoked neurotransmission and highly diminished rate of spontaneous neurotransmitter release (*Bronk et al., 2007*). In Fluo-4 AM imaging experiments with hippocampal cultures made from littermate control mice, the application of AP5 was able to produce a significant decrease in mSCT frequency compared to baseline recorded in TTX, as had been observed previously in wild-type rat cultures. Neurons derived from SNAP25 knock out animals had a significantly decreased baseline mSCT frequency. Also, these transients remained sensitive to AP5, which is again consistent with mSCT generation being driven by spontaneous vesicle release (*Figure 3C,D*).

## mSCTs do not require activation of AMPA receptors, L-type $Ca^{2+}$ channels, or group I mGluRs

Mature glutamatergic synapses contain both AMPA and NMDA receptors (*Bekkers and Stevens, 1989*; *Liao et al., 2001*). Therefore, in the next set of experiments we tested whether concurrent AMPA receptor activity augments NMDA receptor activity at rest through electrical means. Such synergy between the activation of the two types of receptors may be facilitated by dendritic spines that possess a high spine neck resistance that render them electrically isolated from the dendritic shaft (*Bloodgood and Sabatini, 2005*; *Harnett et al., 2012*) but see (*Popovic et al., 2014*). In this way activation of AMPA receptors may result in sufficient local depolarization to facilitate relief of adjacent NMDA receptors from $Mg^{2+}$ block. Additionally, AMPA receptors lacking GluA2, which are calcium-permeable, could also contribute to these transients (*Hollmann et al., 1991*). To examine the role of

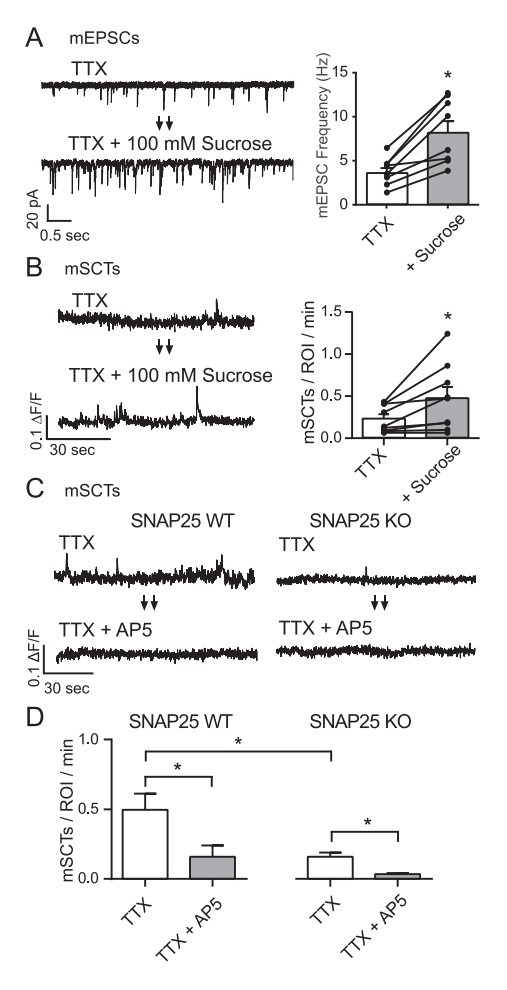

**Figure 3**. mSCT frequency is correlated with mEPSC frequency. (**A**) Whole cell recordings from WT cells (left) show a ~twofold increase in mEPSC frequency when switched to Tyrode's solution containing 100 mM sucrose (right) (p = 0.002, Student's paired T test, N = 8 cells, from 5 coverslips and 2 cultures). (**B**) Example traces (left) and quantification of spontaneous $Ca^{2+}$ transient frequencies measured via imaging show a ~twofold increase upon application of 100 mM sucrose (right) (p = 0.028, Student's paired T test, N = 9 experiments from 3 cultures). (**C**) Fluo-4 example traces from both control and SNAP25 KO animals before and after the application of AP5. (**D**) Fluo-4 imaging in cultures made from SNAP25 KO and littermate control mice reveal that the KO cultures have a substantially decreased mSCT frequency. In this setting, AP5 treatment greatly decreases but does not completely abolish the remaining mSCTs. (WT, TTX vs WT, TTX+AP5 p = 0.010. WT, TTX vs KO, TTX p = 0.008. KO TTX vs KO TTX +AP5 p = 0.003, via 1-way ANOVA with Tukey's multiple comparisons, N = 8 experiments in WT cells and 9 experiments in KO cells from 3 cultures).

AMPA receptor activation on mSCTs, we performed the same analysis above in the presence of AMPA receptor antagonist 2,3-dihydroxy-6-nitro-7-sulfamoyl-benzo[f]quinoxaline-2,3-dione (NBQX). In these experiments NBQX (5 µM) did not affect mSCT frequency (*Figure 4A*). This argues against a direct (e.g., via calcium-permeable GluA2 lacking receptors) or indirect (via local depolarization) contribution of AMPA receptors to mSCTs (*Figure 4A*).

Although experiments presented above showed that the NMDA receptor activity is responsible for triggering the majority of mSCTs in response to spontaneous glutamate release, it remains possible that L-type $Ca^{2+}$ channels may also contribute this activity as they have been shown to open near resting membrane potentials (*Kavalali and Plummer, 1996*; *Magee et al., 1996*). Therefore, we also tested if L-type $Ca^{2+}$ channel activity contributed to the mSCT activity. In these experiments, treatment with the L-type $Ca^{2+}$ blocker nimodipine (5 µM) did not significantly affect mSCT frequency (*Figure 4B*) indicating that these channels do not contribute to the $Ca^{2+}$ transients. However, here we should note that L-type channel activity may still be involved in setting resting $Ca^{2+}$ levels and thus impact signaling (*Wang et al., 2011*). Despite producing no change in mSCT frequency in this assay, nimodipine was able to decrease $Ca^{2+}$ influx in a separate assay (*Figure 4—figure supplement 1*).

In subsequent experiments we also tested the potential role of $G_q$-coupled metabotropic glutamate receptor subtypes 1 and 5 in maintenance of $Ca^{2+}$ transients. These group I mGluRs can affect $Ca^{2+}$ signaling via activation of phospholipase C and IP3 generation (*Skeberdis et al., 2001*; *Topolnik et al., 2006*). However, application of the mGluR1 antagonist YM202074 and mGluR5 antagonist Fenobam did not cause a significant change in mSCT frequency, indicating that activation of these receptors does not contribute to these $Ca^{2+}$ transients (*Figure 4C*). Despite producing no change in the measured mSCT frequency, these drugs were shown to be blocking their target receptors in a separate assay (*Figure 4—figure supplement 2*).

## mSCT generation requires $Ca^{2+}$ release from internal stores

In hippocampal pyramidal cells, NMDA receptor opening by evoked glutamate release elicits larger $Ca^{2+}$ transients through a $Ca^{2+}$-induced $Ca^{2+}$ release mechanism (*Lei et al., 1992*; *Emptage et al., 1999*). In this form of signaling,

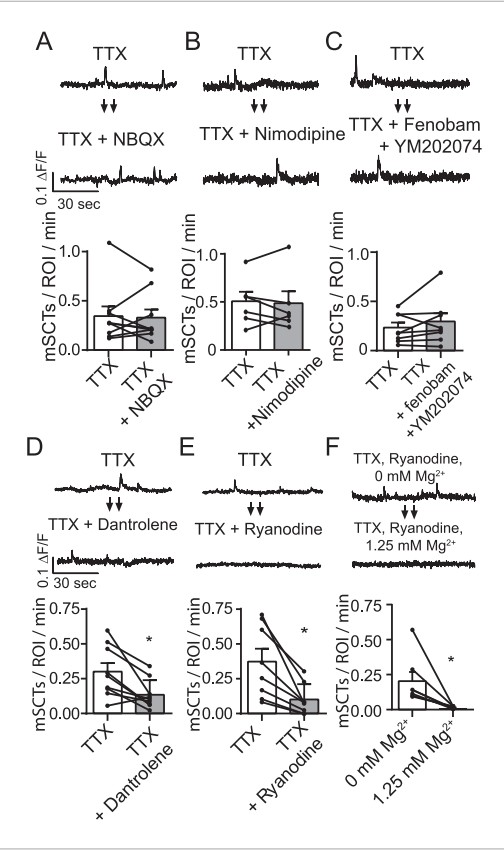

**Figure 4**. Spontaneous Ca²⁺ transient generation is decreased by blocking release of Ca²⁺ release from internal stores but not by blocking the AMPA receptors, L-type Ca²⁺ channels or group I mGluRs. (**A**) Traces (top) from images taken before and after treatment with the AMPA receptor blocker NBQX show no change in Ca²⁺ transient frequency (bottom) (p = 0.78 with Student's paired t-test, N = 9 experiments, 2 cultures). (**B**) Imaging in the presence of the specific L-type calcium channel blocker nimodipine (5 μM) does not affect spontaneous Ca²⁺ transient frequency (bottom) (p = 0.89 with Student's paired t-test. N = 7 experiments 1 culture). (**C**) Imaging in the presence of mGluR1 and mGluR5 blockers YM202074 and fenobam produce no differences in spontaneous Ca²⁺ transient frequency (p = 0.26 via Student's t-test. N = 8 cells, 2 cultures). (**D**) Application of internal Ca²⁺ store blocker dantrolene produces a significant drop in mSCT frequency in before/after experiments. (p = 0.008, Student's paired t-test, N = 9 experiments, 3 cultures). (**E**) Fluo-4 AM example traces and frequency quantification from cells recorded in TTX then in TTX + 30 μM Ryanodine. In before/after imaging experiments, 15 min treatment of the use dependent ER Ca²⁺ channel blocker ryanodine decreased the frequency of observed Ca²⁺ transients (p = 0.004, N = 8 via Student's paired t-test). (**F**) Example traces and frequency quantification of cells pre treated with ryanodine and then imaged first in Tyrode's solution containing no Mg²⁺ followed by Tyrode's solution containing 1.25 mM Mg²⁺. Ca²⁺ transients are

*Figure 4. continued on next page*

the small Ca²⁺ transient produced by the NMDA receptor opening raises internal Ca²⁺ concentrations near ryanodine receptors (RyRs) on the endoplasmic reticulum high enough to cause their opening at which point a much larger transient is generated. To test whether this mechanism plays a role in the mSCTs we observe, we imaged with Fluo-4 AM in the presence of dantrolene, which is known to reduce the Ca²⁺ sensitivity of RyR1 and 3 by blocking their interaction with calmodulin (*Fruen et al., 1997*). Indeed, when compared to the TTX baseline, cells imaged with TTX + dantrolene had a significantly reduced mSCT frequency (*Figure 4D*). This finding was validated by using ryanodine, which directly blocks all three RyR isoforms in a use-dependent manner (*Hawkes et al., 1992*; *Meissner and el-Hashem, 1992*). To facilitate use-dependent block of RyRs by ryanodine, the baseline mSCT frequency was first collected in TTX and then cells were perfused for 13 min with a solution containing 30 μM ryanodine without TTX to maximize RyR opening and ryanodine block. The cells were then perfused with Tyrode's containing TTX + ryanodine for 2 min before continuing the recording. Under these conditions, the application of ryanodine produced a ~fivefold decrease in mSCT frequency (*Figure 4E*). In addition, the Ca²⁺ transients that were detectable after ryanodine treatment were substantially decreased in amplitude suggesting that they are likely to be produced by a sub-population of RyRs that remained unblocked or incompletely blocked (average TTX ΔF/F₀ = 0.052 ± 0.001, TTX + Ryanodine ΔF/F₀ = 0.045 ± 0.001, p = 0.002 Student's unpaired t-test, n = 199 events in TTX, 110 events in TTX + ryanodine, 8 experiments, 2 cultures). Treatment with dantrolene or ryanodine is presumed to decrease mSCT frequency by blocking RyRs responsible for producing the Ca²⁺ transient. With these inhibitors present, further NMDA openings can no longer trigger an mSCT. In fact, the efficacy of ryanodine in this case allowed further investigation of the pure NMDA transient under these experimental conditions. We incubated neurons with ryanodine for 15 min to block RyRs and then loaded them with Fluo-4 AM as before. These cells were imaged in Tyrode's solution containing TTX but no extracellular Mg²⁺ to allow maximal NMDA currents. Under these conditions Ca²⁺ transients were observed, but when 1.25 mM Mg²⁺ was again added no further transients could be measured (*Figure 4F*). These results illustrate that under physiological concentrations of Mg²⁺, Fluo-4 cannot detect the NMDA Ca²⁺ transient without further amplification from Ca²⁺ induced Ca²⁺ release.

*Figure 4. Continued*

measured in Mg²⁺ free solution but not in 1.25 mM Mg²⁺. (p = 0.024, via Student's unpaired t-test. N = 7 experiments 2 cultures).

The following figure supplements are available for figure 4:

**Figure supplement 1**. Nimodipine produces positive results in a separate assay.

**Figure supplement 2**. YM202074 and Fenobam produce positive results in a separate assay.

## Blocking mSCTs induces homeostatic eEF2 kinase-dependent synaptic scaling

In the next set of experiments, we aimed to examine the physiological impact of RyR-dependent mSCTs by focusing on the putative role of these Ca²⁺ signals in regulation of synaptic efficacy. For this purpose, we investigated the role of mSCTs in homeostatic synaptic scaling, which is a compensatory mechanism where neurons scale the strength of their synaptic inputs multiplicatively in a uniform manner in response to global increases or decreases in activity (*Turrigiano et al., 1998*). This response involves the synthesis and insertion of new AMPA receptors and can be strongly induced by blocking both action potentials and NMDA receptors (*Sutton et al., 2004*). Importantly, although synaptic scaling in response to activity blockade occurs within a time frame of 24–48 hr, suppression of resting synaptic activity mediated by spontaneous neurotransmitter release events results in more rapid synaptic scaling detectable within hours (*Sutton et al., 2006*; *Nosyreva et al., 2013*). This suggests that NMDA receptor activation at rest maintains synaptic homeostasis. However, the mechanism by which NMDA receptor activity near resting membrane potentials signals to translation machinery, in particular to eEF2 kinase, has been unclear, especially when one considers the relatively small ion conductance of NMDA receptors at rest due to Mg²⁺ block (*Espinosa and Kavalali, 2009*).

To investigate the role of RyR-dependent mSCTs in homeostatic synaptic scaling, hippocampal neurons were incubated for 3 hr in culture media containing TTX + vehicle (negative control), TTX + ryanodine, or TTX + AP5 as positive control. Neurons were then perfused with Tyrode's solution and whole cell voltage clamp recordings were made in 1 µM TTX, 50 µM PTX and 50 µM AP5 to isolate AMPA-mEPSCs. Under these conditions, the amplitude distributions of AMPA-mEPSCs obtained from neurons treated previously with TTX + ryanodine as well as those treated with TTX + AP5 showed a significant rightward shift towards larger amplitudes compared to the control condition (*Figure 5A,B*). When the collected mEPSC amplitudes were plotted rank order in control vs TTX + ryanodine, a linear fit revealed a scaling factor of 1.28 indicating that cell-wide, mEPSC amplitudes increased uniformly 28% over 3 hr with TTX + ryanodine treatment (*Figure 5C*). This increase in mEPSC amplitudes was not as pronounced as was found with the positive control (TTX + AP5) which may correlate with the finding that ryanodine treatment does not block mSCTs as completely as AP5 (*Figures 2D, 4E*). It is important to note that while other groups have reported an immediate decrease in mEPSC frequency with the acute application of ryanodine (*Emptage et al., 1999*), in our system the mEPSC frequencies in neurons treated with ryanodine for 15 min were indistinguishable from those incubated with vehicle as control (TTX mEPSC freq = 7.59 Hz ± 1.75, TTX + Ryanodine mEPSC freq = 8.92 ± 1.13, p = 0.54 using Student's t-test, N = 7 cells from 5 coverslips, 2 cultures). Since the acute application of ryanodine does not alter mEPSC frequency in this system we believe the synaptic scaling effect mainly results from ryanodine acting at the postsynapse to block mSCT activity.

In earlier experiments homeostatic synaptic scaling that occurs after blockade of resting NMDA receptor activity was shown to rely on protein synthesis, in particular synthesis of new AMPARs rather than the insertion of existing ones (*Sutton et al., 2006*, *2007*). In order to test whether this is the case for RyR block-induced synaptic scaling, we repeated the experiment above with neurons that were treated with the protein synthesis inhibitor anisomycin (20 µM) starting 30 min prior to their 3 hr incubation with TTX. Under these conditions, anisomycin completely abolished the increase in AMPA-mEPSC amplitudes as no significant differences were seen in their distribution after TTX + ryanodine treatment compared to treatment with TTX alone (*Figure 5D–F*).

Previous studies have also shown that a key regulator of protein synthesis, eukaryotic elongation factor 2 (eEF2), is phosphorylated and inactivated by the Ca²⁺-dependent eEF2 kinase thus blocking protein synthesis under resting conditions (*Sutton et al., 2007*; *Autry et al., 2011*; *Nosyreva et al., 2013*;

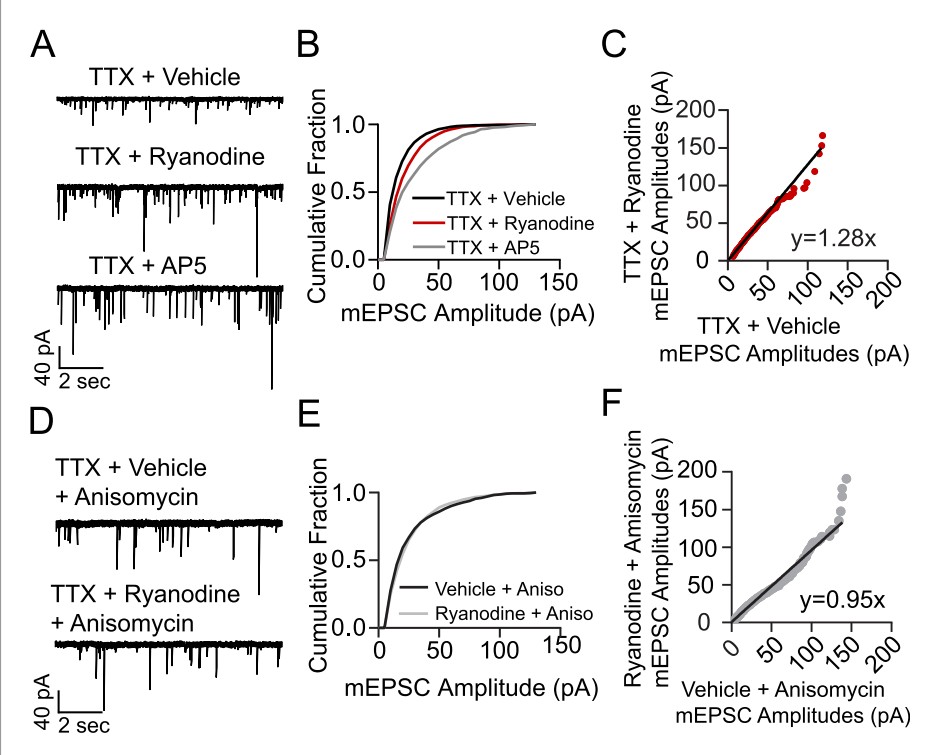

**Figure 5**. Treating cells with ryanodine + TTX produces a protein synthesis dependent increase in mEPSC frequency and amplitude indicative of homeostatic synaptic scaling. (**A**) Example voltage clamp recordings from cells treated with TTX + vehicle (negative control, N = 9 cells from 5 coverslips, 3 cultures), TTX + ryanodine (N = 8 cells from 5 coverslips, 4 cultures) or TTX + AP5 (positive control, N = 6 cells from 4 coverslips, 2 cultures) for 3 hr. (**B**) Cumulative probability histogram showing significant rightward shifts (increases) in the amplitude of AMPA mEPSCs of cells treated with TTX + ryanodine (red line, p = 1.74 × 10$^{-17}$, D = 0.151), or TTX + AP5 (p = 8.79 × 10$^{-40}$, D = 0.255) vs control via Kolmogorov–Smirnov test. (**C**) Rank order plot of TTX + vehicle mEPSC amplitudes vs TTX + ryanodine showing a multiplicative scaling factor of 1.28. (**D**) Example voltage clamp recordings from cells pretreated for 30 min with the protein synthesis inhibitor anisomycin and then TTX + vehicle (N = 6 cells from 5 coverslips, 4 cultures) or TTX + ryanodine for 3 hr (N = 7 cells from 5 coverslips, 2 cultures). (**E**) Cumulative probability histogram of mEPSC amplitudes shows no significant difference between treatment groups when pretreated with anisomycin (p = 0.078, D = 0.052 via Kolmogorov–Smirnov test). (**F**) Rank order plot of mEPSC amplitudes indicates that anisomycin pretreatment abolishes scaling between treatment groups.

*Gideons et al., 2014*). To test whether RyR-mediated mSCTs could be tonically activating eEF2 kinase and thus inhibiting protein synthesis in dendrites, we tested the impact of ryanodine treatment in hippocampal neuronal cultures from eEF2 kinase knockout mice. In hippocampal neurons made from wild-type littermate controls, treating with TTX + ryanodine for 3 hr produced a significant increase in mEPSC amplitudes compared to TTX + vehicle, where plotting the amplitudes in rank order revealed a 42% increase in synaptic strength (*Figure 6A–C*). When the same experiment was performed using neurons from eEF2 kinase knockout mice, treatment with TTX + ryanodine did not produce a significant shift in mEPSC amplitudes (*Figure 6D,E*). The rank order plot revealed only a 1% difference in synaptic strength between treatment groups (*Figure 6F*). Taken together these results suggest that RyR-dependent mSCT-driven signaling acts through Ca$^{2+}$-dependent eEF2 kinase to maintain synaptic homeostasis.

## Discussion

In this study, we took advantage of multiple Ca$^{2+}$ indicator probes to examine the properties of Ca$^{2+}$ transients detected in hippocampal neurons in physiological levels of extracellular Mg$^{2+}$ in the absence of action potentials. These transients are important because they are key to understanding the Ca$^{2+}$ signaling events occurring at rest that result in regulation of protein translation and gene

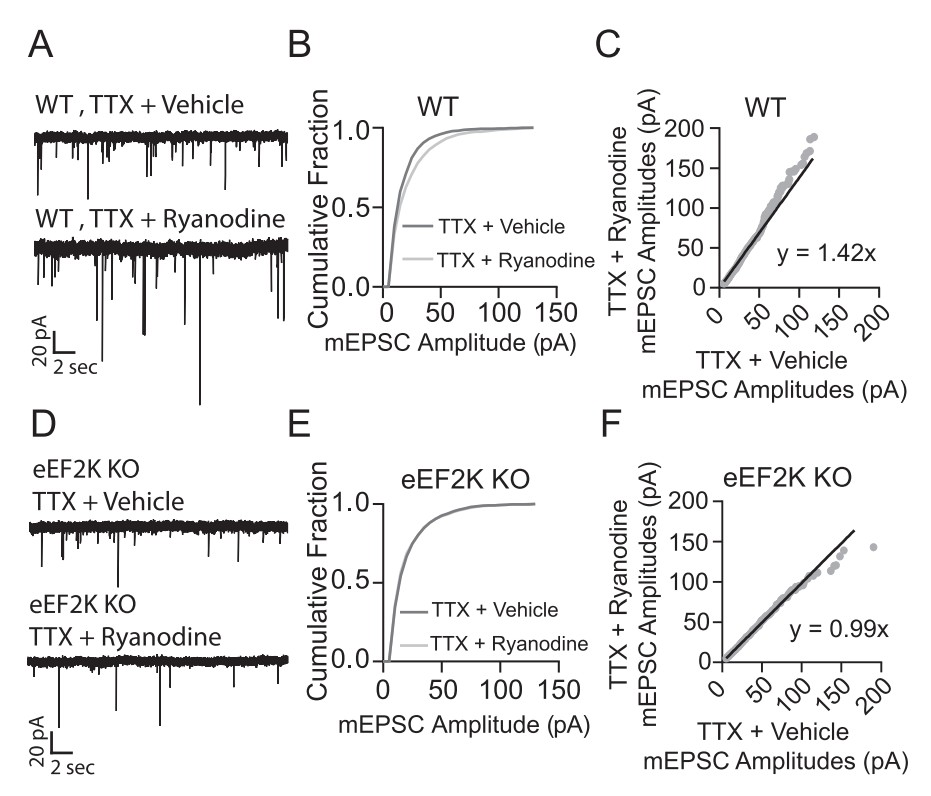

**Figure 6**. Ryanodine treatment does not trigger homeostatic synaptic scaling in eEF2 kinase knockout neurons. (**A**) Example traces from WT littermate mice with and without 3 hr ryanodine treatment. (**B**) Cumulative probability histogram shows a significant shift in mEPSC amplitude in TTX + ryanodine treated animals (N = 12 cells, from 7 coverslips, 3 cultures) vs TTX + vehicle control (N = 13 cells from 8 coverslips, 3 cultures) (p = 3.32 × 10$^{-14}$, D = 0.112 via Kolmogorov–Smirnov test). (**C**) Rank order plot shows a 1.42 fold increase in synaptic strength after ryanodine treatment. (**D**) Example traces from eEF2 kinase KO animals with and without ryanodine treatment. (**E**) Cumulative probability histogram shows no shift in the distribution of mEPSC amplitudes in eEF2K KO animals treated with TTX + ryanodine (N = 9 cells, from 6 coverslips, 3 cultures) vs TTX + vehicle (N = 13 cells from 7 coverslips, 3 cultures) (p = 0.066, D = 0.058 via Kolmogorov–Smirnov test). (**F**) Rank order plot shows no appreciable multiplicative change in synaptic strength in the eEF2 K KO animals with ryanodine treatment.

transcription leading to synaptic plasticity (*Chen et al., 2014*; *Lalonde et al., 2014*) (for review see *Kavalali, 2015*). Under these conditions we detected robust NMDA receptor dependent Ca$^{2+}$ transients at a rate of 0.32 ± 0.03 min$^{-1}$ where previously our group was able to measure a per synapse spontaneous release rate of 0.76 ± 0.03 min$^{-1}$ by imaging with presynaptic probes (*Leitz and Kavalali, 2014*). The relatively higher release rate measured earlier may suggest that not every release event is able to generate an mSCT. The link between these Ca$^{2+}$ transients and spontaneous neurotransmitter release was verified by the parallel increase in mSCT and spontaneous neurotransmitter release frequencies in response to application of hypertonic sucrose. Furthermore, the frequency of Ca$^{2+}$ transients was also significantly diminished in neurons lacking SNAP-25, which show a substantial reduction in spontaneous release, and in cultures treated with NMDA receptor blockers.

Interestingly GCaMP5K-PSD95, a probe located near the postsynaptic density, revealed only a very small number of events that fulfilled our detection criteria while soluble probes proved to be much better indicators. This observation indicates that although mSCT generation depends on NMDA receptor driven Ca$^{2+}$ influx, this does not result in strong signals at the postsynaptic density. Rather, mSCTs seem to rely on the activation of Ca$^{2+}$ release from smooth endoplasmic reticulum which is present in spines or adjacent dendritic regions (*Spacek and Harris, 1997*).

The generation of mSCTs was not dependent on AMPA receptors, L-type $Ca^{2+}$ channels or postsynaptic metabotropic glutamate receptor subtypes 1 and 5. While all of these play a role in $Ca^{2+}$ dynamics under other circumstances the mSCTs we observed under resting conditions were primarily driven by the coupling of the NMDA receptor to internal $Ca^{2+}$ stores through the ryanodine receptor, as the application of dantrolene or ryanodine produced a marked reduction in both mSCT frequency and amplitude. Earlier studies performed in hippocampal synapses discovered that unitary evoked EPSCs were accompanied by $Ca^{2+}$ transients that were only minimally dependent on voltage gated $Ca^{2+}$ channels or AMPA receptors. However, unlike the mSCTs we observe, the application of ryanodine produced only a small reduction in $Ca^{2+}$ transient amplitude in these experiments (*Kovalchuk et al., 2000*). This difference may suggest that spontaneous glutamate release-driven $Ca^{2+}$ transients are more dependent on internal $Ca^{2+}$ stores compared to $Ca^{2+}$ transients elicited by evoked release.

In this study, we tested a key prediction of these observations on synaptic plasticity by assessing the role of $Ca^{2+}$-induced $Ca^{2+}$ release in synaptic scaling triggered at rest. Our experiments showed that the synaptic scaling produced by the blockade of spontaneous NMDA-mEPSCs is also produced by blocking the $Ca^{2+}$ release from internal stores indicating a strong link between the two signals. The generation of relatively large store-driven $Ca^{2+}$ transients provides a critical amplification step for the relatively small NMDA-mEPSCs seen under physiological conditions (*Espinosa and Kavalali, 2009*; *Gideons et al., 2014*). The resulting signal is delocalized and pulsatile which may allow synaptic NMDA receptors to exert signaling influence in the surrounding dendritic regions. This could be critical for local translational control as eEF2 localizes to the dendritic shaft rather than dendritic spines (*Asaki et al., 2003*). The low frequency of observed mSCTs may also be a defining attribute, as the ubiquitous $Ca^{2+}$ binding protein calmodulin is predicted to interact with different target kinases and enzymes based on the frequency and duration of its activation by free $Ca^{2+}$ (*Saucerman and Bers, 2008*; *Slavov et al., 2013*). Taken together these findings identify a critical missing mechanistic link between spontaneous neurotransmission and the control of dendritic signaling events that regulate synaptic efficacy.

## Materials and methods

### Cell culture

Hippocampal cultures from Sprague–Dawley rats or eEF2 kinase knockout mice and their wild-type littermate controls were generated from postnatal day 1–3 male and female pups and plated on Matrigel (Corning Inc, NY) coated coverslips as described previously (*Kavalali et al., 1999*). Neurons were infected with lentivirus at 4 days in vitro. Neurons were used for experiments between 14 to 18 days in vitro.

Dissociated hippocampal cultures from SNAP25 knockout mice and their wild-type littermates were generated from E17-20 embryos and were plated on poly-d-lysine coated coverslips as described previously (*Bronk et al., 2007*). Neurons were used for experiments 14–18 days in vitro.

### Whole cell voltage clamp recordings

Dissociated hippocampal cultures aged 14–18 days in vitro were voltage clamped at −70 mV using an Axon Instruments Axopatch 200B amplifier with access resistances less than 25 MΩ for each recording. Internal pipette solution contained (in mM): 120 K-Gluconate, 20 KCl, 10 NaCl, 10 HEPES, 0.6 EGTA, 4 Mg-ATP and 0.3 Na-GTP at pH 7.3. To isolate AMPA-mEPSCs, the extracellular solution contained 1 µM TTX, 50 µM picrotoxin (PTX, to block mIPSCs) and 50 µM (2*R*)-amino-5-phosphonovaleric acid (AP5), 2 mM $Ca^{2+}$ and 1.25 mM $Mg^{2+}$. All whole cell patch clamp recordings were performed under continuous perfusion. Cells were perfused for 3-min prior to recording to achieve stable baselines. No more than 2 recordings were obtained per coverslip.

### Whole cell voltage clamp statistics and analysis

AMPA-mEPSCs were quantified using Synaptosoft MiniAnalysis software. Frequency data was collected by quantifying 4 min per cell starting at the beginning of each recording. To ensure that high

frequency cells did not skew the amplitude comparisons by being over represented, 200 mEPSC amplitudes were randomly selected from each recording to build the cumulative probability histograms and rank order plots. Kolmogorov–Smirnov test was performed using Past 3.02 (http://folk.uio.no/ohammer/past/).

## Ca$^{2+}$ imaging

### Fluo-4 AM imaging
Neurons were incubated for 10 min in culture media containing 5.6 µM Fluo-4 AM (Life Technologies, Grand Island NY). Coverslips were then removed and washed for 2 min prior to recording with Tyrode's solution containing (in mM) 150 NaCl, 4 KCl, 1.25 MgCl$_2$, 2 CaCl$_2$ and 10 TES buffer, pH adjusted to 7.4. Where solutions are noted to have 0 mM Ca$^+$ the solution is prepared from deionized water with no added Ca$^{2+}$. All solution changes except +100 mM hypertonic sucrose include 2 min of wash time in the new solution to ensure full application of the new conditions. Neurons were imaged using a 40× objective on a Nikon TE2000-U microscope. Images were collected at 10 frames per second using an Andor xION Ultra EMCCD camera for a duration of ∼ 2 min. Event frequencies per ROI were estimated using the population average obtained from 72 ROIs monitored per experiment. Our analysis, therefore, refers only to average per ROI frequency per experiment. Illumination was provided by a Sutter DG-4 arc lamp using a 470 ± 40 nm bandpass excitation filter. Post experiment synapse visualization used a 548 ± 10 nm filter to excite Syb2-mO and Tyrode's solution containing 50 mM NH$_4$Cl to maximize fluorescence. The emission filter in place allowed 515 ± 15 nm and 590 ± 20 nm bands to pass. Fluo-4 traces were generated by measuring circular ROI, 3 µm radius centered over Syb2-mO puncta.

### Single cell Fluo-4 imaging
Uninfected neurons were loaded with indicator by using the whole cell voltage clamp configuration described above. The patch pipette contained 200 µM Fluo-4 pentapotassium salt (Life Technologies, Grand Island NY).

### Soluble GCaMP5K imaging
Wild type neurons expressing Syb2-mO were transfected with pFU-GCaMP5K using lipofectamine 3000 (Life Technologies, Grand Island NY) 8 hr prior to imaging. Images were collected as above.

### GCaMP5K-PSD95 imaging
Neurons expressing GCaMP5K-PSD95 via lentiviral infection were imaged as indicated above except using a 100× objective. Images were collected at 8 frames per second to minimize single frame noise.

### Ca$^{2+}$ transient analysis and statistics
Ca$^{2+}$ transient frequency was derived from imaging traces by counting Ca$^{2+}$ transients where the signal peak had a 2-point slope greater than 70 A.U. (350 units/s over a 200 ms window) and amplitude greater than 0.035 $\Delta F/F_o$. Ca$^{2+}$ signals were not counted if their peak width was greater than 5 s. Maximum peak amplitude was required to be 2 standard deviations greater than the mean of the signal in the previous 2 s. Single high points were not counted. Detected peaks were ignored if another peak was detected in the following 400 ms to prevent the double counting of slower mSCTs. All error bars represent standard error of the mean except in *Figure 1A–D* where standard deviation is used. Rise and decay times are displayed as τ, where τ is the time in seconds necessary to reach $\left(1-\frac{1}{e}\right)\Delta F$ for the rising phase or $\left(\frac{1}{e}\right)\Delta F$ for the decay phase based on a single exponential fit line obtained with Axon Clampfit 9.0.1.07. All statistical tests were performed using Graphpad Prism 6.01.

## Acknowledgements
We thank members of the Kavalali and Monteggia laboratories, in particular Dr Devon Crawford and Erinn Gideons for insightful discussions and comments on the manuscript. We would also like to thank Tom Reese for his assistance in streamlining the data analysis. This work was supported by NIH grants MH066198 (ETK) and the Cellular Biophysics of the Neuron Training Program T32 NS069562 (ALR).

# Additional information

## Funding

| Funder | Grant reference | Author |
|---|---|---|
| National Institutes of Health (NIH) | MH066198 | Ege T Kavalali |
| National Institute of Neurological Disorders and Stroke (NINDS) | T32 NS069562 | Austin L Reese |

The funders had no role in study design, data collection and interpretation, or the decision to submit the work for publication.

## Author contributions

ALR, Conception and design, Acquisition of data, Analysis and interpretation of data, Drafting or revising the article; ETK, Conception and design, Analysis and interpretation of data, Drafting or revising the article

## Ethics

Animal experimentation: This study was performed in strict accordance with the recommendations in the Guide for the Care and Use of Laboratory Animals of the National Institutes of Health. All of the animals were handled according to approved institutional animal care and use committee (IACUC) protocols of the UT Southwestern Medical Center (APN# 0866-06-05-1).

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
