## [Decision Letter]

Thank you for submitting your work entitled “Spontaneous neurotransmission signals through store-driven Ca^2+^ transients to maintain synaptic homeostasis” for further consideration at *eLife*. Your article has been favorably evaluated by Eve Marder (Senior editor), a Reviewing editor, and three reviewers. The manuscript has been greatly improved over your previous submission but there are two minor corrections that must be addressed before acceptance, regarding apparent inconsistencies/ambiguities in the text. The first may be a typo regarding AP5 sensitivity reported in text vs. legend, and the second has to do with quantification of a low rate from relatively short recordings (see details below).

In general the reviewers were convinced by the revisions. Two reviewers said: “I am satisfied with all of the revisions and hope to see the paper published” and “The authors have addressed all my concerns reasonably. I have no further concerns.”

The third writes: “Most of my concerns were satisfactorily addressed but two points remain.” That reviewer listed the two inconsistencies:

1) There is an apparent contradiction between the text in subsection “mSCTs are driven by spontaneous neurotransmitter release” and in the legend to Figure 3. Text: “Neurons derived from SNAP25 knock out animals had a significantly decreased baseline mSCT frequency. Also, these transients remained sensitive to AP5, which is again consistent with mSCT generation being driven by spontaneous vesicle release (Figure 3).”

The statement “these transients remained sensitive to AP5” seems to be contradicted by the statement in the legend: “The remaining mSCTs in the KO animals are unaffected by AP5.” Is “unaffected” incorrect in the legend? This needs to be clarified in the manuscript.

2) The recordings were 2 minutes long. In the third paragraph of the subsection “Visualization of miniature spontaneous Ca^2+^ transients in hippocampal neurons” it states an mSCT frequency of 0.009 per min per ROI. It needs to be explained how such a low frequency (∼0.5 per hour) can be obtained from 2 min long recordings.

[Editors’ note: a previous version of this study was rejected after peer review, but the authors submitted for reconsideration. The previous decision letter after peer review is shown below.]

Thank you for choosing to send your work entitled “Spontaneous neurotransmission signals through store-driven Ca^2+^ transients to maintain synaptic homeostasis” for consideration at *eLife*. Your full submission has been evaluated by Eve Marder (Senior editor), a Reviewing editor, and three peer reviewers, and the decision was reached after discussions between the reviewers. Based on our discussions and the individual reviews below, we regret to inform you that your work cannot be accepted in its present form for publication in *eLife*. The reviewers made specific suggestions for improvement of the manuscript, however, and if you wish to do the necessary experiments and carry out the suggested revisions, we would be willing to consider a new version of the manuscript. This decision was made not because we didn't find the work potentially exciting, but because the experiments that the reviewers thought important might take more than several weeks to accomplish, and it is against e*Life* policy to ask for extensive new experiments for a revision.

In the discussion of the reviews, three major topics emerged, which all reviewers generally agreed upon. The first is the need for a better description of the analysis, which will likely require some modification of the analysis. Ideally these would resolve the inconsistencies noted in the first review. The second has to do with the N-values and the validity of the statistics. Not only were N-values low, but explicit indication of what “N” refers to in the experiments would also be appropriate (how many independent cultures, how many different coverslips, whether similar #'s of mini's collected and considered from each cell, etc). Third were experimental questions about results that appear contradictory (why ryanodine affects Ca transient frequency but only affects mEPSC amplitude) or ambiguous (in Figure 1, what is the relative sensitivity of the two different Ca^2+^ indicators, and why it is valid to directly compare data obtained using those different constructs).

The complete reviews are included below to facilitate a new submission, should you decide to submit one. Please note that it would likely require some additional experiments to address the concerns of low N as well as rewriting and reanalysis.

Reviewer #1:

In this manuscript the author present experiments investigating miniature spontaneous Ca^2^+ transients (mSCTs) in hippocampal neurons. They report that these are activated by spontaneous glutamate release events and enhanced by Ca^2+^-induced Ca^2+^release, confirming previous results. New insight comes mainly from Figures 5 and 6. The authors show that inhibition of Ca^2+^-induced Ca^2+^ release increases mEPSC amplitudes, indicative of synaptic scaling. This increase is abolished by a protein synthesis inhibitor and in neurons form eEF2 kinase knock-out mice, indicating that ryanodine induced synaptic scaling acts via eEF2 kinase activation.

While there is some interesting new insight from this manuscript, much of it largely confirms previous results, several sections are very poorly written and a number of technical questions arise. In particular, the description of mSCT detection and analysis is inconsistent, raising serious questions regarding the significance of observed phenomena.

Specific points:

1) The event detection/identification needs clarification (Discussion and second paragraph, Figure 2). For the 2 SD threshold 5% of points will exceed the threshold. The Methods section gives only criteria of maximum duration. Was there a minimum duration?

2) How were rise times and decay times defined and determined?

3) The average/median amplitudes of ∆F/F=0.054/0.048 seem to be nearly equal to 2 SD in Figure 2.

4) The legend of Figure 2 states that events having “peak ∆F/F greater than 0.04 were counted if the peak fluorescence value was 2 standard deviations greater than the mean fluorescence”. Methods, subsection “CA^2+^ transient analysis and statistics” states 0.035, and the distribution in Figure 2 (right) shows counts for amplitudes even smaller. The analysis seems inconsistent.

5) According to the Methods section Fluo-4 and GCaMP5k were excited with 470***±*** 40 nm filter, which will also excite Syb2-mOrange, which will change it fluorescence upon vesicle fusion. What emission filter was used to exclude crosstalk?

6) For Figure 1 ROIs were selected over Syb2mOrange puncta determined after NH_4_Cl addition. What was the size and shape of the ROIs?

7) Introducing Fluo-4 salt with a patch pipette reduced observation frequency of mSCTs and the authors interpret this as being due to wash-out. However, it seems that the pipette solution contained high gluconate, which is known to buffer Ca^2+^ (*K*_d_∼35 mM). Although *K*_d_ may appear high it would be expected to affect [Ca^2+^] due to its very high concentration. Washout should produce a decrease in frequency over time compared to Fluo-4 AM, was this observed?

8) Frequency of mSCTs measured with GCaMP5k-PSD95 was ∼1 very 2 hours. How long were the recordings?

9) In the fourth paragraph of the subsection “Visualization of miniature spontaneous CA^2+^ transients in hippocampal neurons” and in Figure 1: The authors claim that differences were insignificant except comparing GCaMP5k-PSD to Fluo-4 AM. From the error bars in Figure 1 this seems not to be supported by the data.

10) Does the lack of events with GCaMP5k-PSD indicate that these are extrasynaptic? If so, this should be stated more clearly.

11) Figure 4: For these negative results – how did the authors ensure that the blockers actually worked in these experiments effectively blocking the respective receptors and L-type Ca current?

12) The legend of Figure 4: “The remaining mSCTs in the KO animals are unaffected by AP-5”. This seems not to refer to the data shown in Figure 4.

13) Figure 5 shows a much larger effect of AP5 than ryanodine. How do the authors interpret this result?

Reviewer #2:

This is an interesting and timely paper on a topic that has received much recent attention in neuroscience. I believe that this paper should be accepted, but there is one somewhat large issue regarding sample size that need be addressed first:

Throughout the paper, the sample size for the experiments is low (N of 5 to 6 is not uncommon). For example, in Figure 4, which encompasses five separate experiments, the total N for all five experiments appears to be less than 30.

(This general issue, RE sample size and statistics in neuroscience, is discussed here: http://www.nature.com/nrn/journal/v14/n5/full/nrn3475.html). It seems that the N should be increased in some experiments reported here.

In the first paragraph of the subsection “Visualization of miniature spontaneous CA^2+^ transients in hippocampal neurons”: The frequency of spontaneous Ca^2+^ transients was 0.32 per minute. If you sum the Ca^2+^ transients across all release sites in entire cell, do you get a Ca^2+^ transient frequency that roughly matches the mEPSC frequency (∼1Hz)? If so, that would be a good validation of the method. If not, then either there is a problem with the method or there's some fraction of release that does not trigger NMDAR/CICR signaling, which would be good to know. If this is not technically feasible, please explain.

Figure 5: It is surprising that ryanodine had no effect on mEPSC frequency. From the representative traces, it would appear that ryanodine did have an effect. Maybe these are not very representative? Or, do the authors think there was a slight effect of ryanodine in mEPSC frequency that did not reach significance? Please address.

Reviewer #3:

In this manuscript Reese and Kavalali report how spontaneous neurotransmission influences synaptic homeostasis. They show action potential independent calcium transients in the dendritic region using multiple calcium indicators. The calcium transients required NMDA receptor activity and calcium release from internal stores. Furthermore they show that reducing the calcium transient through inhibiting NMDA receptors or blocking internal stores results in synaptic scaling mediated by protein synthesis and eukaryotic elongation factor 2 (eEF2). The authors suggest that spontaneous neurotransmission through NMDA receptor driven calcium transient is required for synaptic homeostasis.

The manuscript has several interesting findings, however one major issue is that there is quite a jump between the story on what is the source of spontaneous dendritic calcium transients to how it affects the synapse. Either they need to provide better transition or posit the paper in a different manner. Figure 1 through 4 make sense together, but the last two figures, which are addressing synaptic scaling, appear without much of a transition.

1) In Figure 1, the calcium transient from Fluo-4 AM is not increased in 0 magnesium, the authors do point that out. In light of what is indicated in the Introduction, where they state '…that calcium signal is 20% of the full calcium signal of unblocked NMDA…', it is quite surprising that the signal is not higher in 0 magnesium. This needs to be addressed. Furthermore, because Fluo-4 AM detects calcium events both in the presynapse and postsynapse, an explanation or experiment is required to distinguish these two sources. How did they define the calcium events as dendritic only? Figure 1, is the impermeable Fluo-4 experiment conducted in Syb2-mO infected neurons? If so, it is worth checking if there is any spontaneous transients in the presynaptic terminal of cells that express Syb2-mO.

2) The authors showed mSCT frequency is nicely correlated with mEPSC frequency, when influenced by presynaptic players (Figure 3). On the other hand intracellular calcium store blockers, Dantrolene and Ryanodine potentially only affecting the dendrite each reduce mSCT frequency (Figure 4). It is surprising and also hard to understand how calcium release from internal stores affects frequency rather than amplitude or kinetics of the calcium transients. It would be important to address if the presynapse is affected by Dantrolene and Ryanodine.

3) Figure 5, the authors indicate that ryanodine and AP5 each only influence mEPSC amplitudes not the frequency. From the sample trace it appears that frequency is actually increased in both cases, a more representative trace is needed here.

4) Why does ryanodine affect the calcium transient frequency, but affects only the mEPSC amplitude? This seems contradictory. I would expect an effect on the mEPSC frequency as well.

---

## [Author Response]

*1) There is an apparent contradiction between the text in subsection “mSCTs are driven by spontaneous neurotransmitter release” and the legend to*
Figure 3*. Text: “Neurons derived from SNAP25 knock out animals had a significantly decreased baseline mSCT frequency. Also, these transients remained sensitive to AP5, which is again consistent with mSCT generation being driven by spontaneous vesicle release (*Figure 3*)*.*”*

*The statement* “*these transients remained sensitive to AP5*” *seems to be contradicted by the statement in the legend:* “*The remaining mSCTs in the KO animals are unaffected by AP5.*” *Is* “*unaffected*” *incorrect in the legend? This needs to be clarified in the manuscript.*

What is meant is that the overall population of mSCTs in the knockout animals is sensitive to AP5, but after AP5 treatment there are some mSCTs that remain unaffected. The legend for Figure 3 has been edited and now reads:

*“*D. Fluo-4 imaging in cultures made from SNAP25 KO and littermate control mice reveal that the KO cultures have a substantially decreased mSCT frequency. In this setting, AP5 treatment greatly decreases but does not completely abolish the remaining mSCTs. (WT, TTX vs WT, TTX+AP5 p = 0.010. WT, TTX vs KO, TTX p = 0.008. KO TTX vs KO TTX+AP5 p = 0.003, via 1-way ANOVA with Tukey’s multiple comparisons, N=8 experiments in WT cells and 9 experiments in KO cells from 3 cultures).*”*

*2) The recordings were 2 minutes long. In the third paragraph of the subsection “Visualization of miniature spontaneous Ca*^*2+*^
*transients in hippocampal neurons” it states an mSCT frequency of 0.009 per min per ROI. It needs to be explained how such a low frequency (∼0.5 per hour) can be obtained from 2 min long recordings.*

Recordings were 2 minutes long to prevent photodamage and bleaching, which would have resulted in less reliable data. However, in each experiment we measured around 72 ROIs and estimated event frequencies per ROI using the population average (equivalent to ∼ 2 hrs per experiment). In fact, for this reason we do not report per synapse frequency distributions in the manuscript and only refer to average per ROI frequency per experiment. This point is now clarified in the Methods section:

**“**Event frequencies per ROI were estimated using the population average obtained from 72 ROIs monitored per experiment.”

[Editors’ note: the author responses to the previous round of peer review follow.]

In the discussion of the reviews, three major topics emerged, which all reviewers generally agreed upon. The first is the need for a better description of the analysis, which will likely require some modification of the analysis. Ideally these would resolve the inconsistencies noted in the first review.

We followed the reviewers’ advice and now expanded the description of our analysis.

*The second has to do with the N-values and the validity of the statistics. Not only were N-values low, but explicit indication of what* “*N*” *refers to in the experiments would also be appropriate (how many independent cultures, how many different coverslips, whether similar #'s of mini's collected and considered from each cell, etc).*

N-values are now clearly stated in each figure legend as number of coverslips or cells and the number of cultures. In several cases we conducted new experiments to increase N-values.

*Third were experimental questions about results that appear contradictory (why ryanodine affects Ca transient frequency but only affects mEPSC amplitude) or ambiguous (in*
Figure 1*, what is the relative sensitivity of the two different Ca*^*2+*^
*indicators, and why it is valid to directly compare data obtained using those different constructs).*

We now clarified these issues in the manuscript. Briefly, mEPSCs report synaptic AMPA receptor activity, whereas Ca^2+^ transients reflect RyR mediated Ca^2+^ release events triggered by residual NMDA receptor activity (seen in the presence of Mg^2+^). Therefore, the two measures, although both dependent on spontaneous glutamate release and occur concurrently, report distinct nodes in the signaling cascade. Please also note that in our results Ca^2+^ transients are not sensitive to AMPA blockers in contrast to their extreme sensitivity to NMDA receptor block (Figure 2).

Overall, in revising this manuscript, we have done the following:

1) We have clarified the description for the signal processing and analysis of Fluo-4 AM calcium transients as well as verifying that all measurements included in this manuscript conform to these analysis standards. We have included in this clarification a more detailed Figure 2 that illustrates the process used in detecting these calcium signals.

2) We have performed new measurements to increase N numbers throughout. New experiments have been added to the experiments in Figure 2 so that the final N number is 8.

3) New experiments have been performed to add to Figure 3 so that the final N numbers are 8 and 9 respectively. This new data increases the confidence of our findings by further decreasing the p values to 0.002 and 0.028 respectively.

4) New experiments have been performed to add to panels 4C, 4D and 4E. These experiments bolster the N numbers to 8, 9 and 8 respectively.

A new experiment has been performed to clarify the makeup of the calcium transients detected with Fluo-4 AM. This experiment is now displayed as Figure 4. Briefly, ryanodine treated cells (ER store calcium is blocked) were recorded in Fluo-4 AM with and without magnesium in order to confer or relieve the voltage dependence of the NMDA receptor. With no magnesium in the bath the undamped calcium transient of the NMDA receptor can be measured but when magnesium is added to the bath solution no further transients can be measured. This experiment illustrates that the calcium transients measured in 1.25 mM Mg^2+^ with Fluo-4 AM include a negligible amount of signal from the NMDA receptor itself and are primarily driven by ER stores.

6) A control experiment has been done to illustrate that under these recording conditions, no appreciable signal from the red channel (Synaptobrevin2-mOrange) is bleeding into the green channel (Fluo-4 AM). Under the conditions used throughout the paper to detect calcium transients with Fluo-4 AM, no appreciable bleed through was detected between the two channels.

7) Control experiments were performed to confirm the efficacy of Nimodipine. The same stock solution that was previously used in the manuscript was again tested and found to be working in a separate assay.

Control experiments were performed to confirm the efficacy of YM202074 and Fenobam. The same stock solutions that were previously used in the manuscript was again tested and confirmed to be working in a separate assay.

Reviewer #1:

*In this manuscript the author present experiments investigating miniature spontaneous Ca*^*2*^*+ transients (mSCTs) in hippocampal neurons. They report that these are activated by spontaneous glutamate release events and enhanced by Ca*^*2+*^*-induced Ca*^*2+*^*release, confirming previous results. New insight comes mainly from*
Figures 5 and 6*. The authors show that inhibition of Ca*^*2+*^*-induced Ca*^*2+*^
*release increases mEPSC amplitudes, indicative of synaptic scaling. This increase is abolished by a protein synthesis inhibitor and in neurons form eEF2 kinase knock-out mice, indicating that ryanodine induced synaptic scaling acts via eEF2 kinase activation.*

While there is some interesting new insight from this manuscript, much of it largely confirms previous results, several sections are very poorly written and a number of technical questions arise. In particular, the description of mSCT detection and analysis is inconsistent, raising serious questions regarding the significance of observed phenomena.

We thank the reviewer for his/her constructive criticisms. We respectfully disagree that our findings are largely confirmatory. We believe that our results provide insight into a critical open question in neuroscience, namely how spontaneous glutamate release events manage to regulate neuronal signaling and synaptic efficacy. We addressed the technical issues raised by the reviewer as follows:

Specific points:

*1) Discussion, second paragraph,*
Figure 2*: The event detection/identification needs clarification. For the 2 SD threshold 5% of points will exceed the threshold. The Methods section gives only criteria of maximum duration. Was there a minimum duration?*

Detail has been added to the Methods section. The minimum duration for an event was 200 ms as single high points were not counted. However, very few detected events were this fast as you can see from the decay times shown in Figure 2.

2) How were rise times and decay times defined and determined?

The Materials and methods as well as the Figure 2 legend have been updated.

Briefly, the decay time (τ) is the time that it takes the signal to decrease to 1/e times the peak value (∼37% of the peak). The rise time (τ) is the time that it takes for the signal to reach 1/1-e times the peak value from baseline (∼63% of the max value).

*3) The average/median amplitudes of ∆F/F=0.054/0.048 seem to be nearly equal to 2 SD in*
Figure 2*.*

We now better illustrate our selection criteria in Figure 2. The analysis requires that when the two point slope exceeds 70 (350 A.U./sec) and the ∆F/Fₒ of the second point exceeds 0.035, the amplitude of the second point must exceed 2 SD greater than the mean of the signal 2 seconds previous to be counted as a peak. In Figure 2, the ∆F/Fₒ threshold is represented by the red line and the gray shaded region shows the area covered by 2SD +/-the mean. For comparison, the bottom red trace shows the two point slope (first derivative approximation) and the 350 A.U/sec threshold.

*4) The legend of*
Figure 2
*states that events having* “*peak ∆F/F greater than 0.04 were counted if the peak fluorescence value was 2 standard deviations greater than the mean fluorescence*”*. Methods, subsection “CA*^*2+*^
*transient analysis and statistics” states 0.035, and the distribution in*
Figure 2
*(right) shows counts for amplitudes even smaller. The analysis seems inconsistent.*

We apologize for this oversight. We now corrected this error. The actual value we used in our analysis is 0.035 (not 0.04). Figure 2 has been re-analyzed and now includes amplitude data taken directly from the analysis files.

5) According to the Methods section Fluo-4 and GCaMP5k were excited with 470± 40 nm filter, which will also excite Syb2-mOrange, which will change it fluorescence upon vesicle fusion. What emission filter was used to exclude crosstalk?

We thank the reviewer for pointing out this important issue. We have two answers to this question:

A) A dichroic filter is in place that allows both wavelengths to pass. This information is now included in Methods. The dichroic emission filter allows 515 ± 15 nm and 590 ± 20 nm bands to pass. This configuration allows the emission wavelengths for both mOrange and Fluo-4 to pass.

In our hands, synaptobrevin-mOrange (unlike synaptophysin-pHtomato or vGlut-pHluorin see [25]) is not suited to single vesicle imaging due to its poor signal to noise ratio and the rapid lateral diffusion of the synaptobrevin probe. To illustrate this, dual channel imaging was performed alternating between Fluo-4 AM and Syb2-mO excitation. Images were collected at the same exposure and gain settings as all other Fluo-4 AM experiments (100 ms exposure). 382 peaks were detected in the fluo-4 channel and aligned by peak value to generate an average event. The corresponding Syb2-mO signal was also averaged, but no signal is detected (see Figure 2—figure supplement 1).

*6) For*
Figure 1
*ROIs were selected over Syb2mOrange puncta determined after NH*_*4*_*Cl addition. What was the size and shape of the ROIs?*

ROIs were in 3 µm radius around the syb2-mOrange puncta. We now included this information in the Methods section. The need for larger ROIs was necessitated by the fact that mSCTs (seen in the presence of Mg^2+^) typically occur within the vicinity but not directly juxtaposed to presynaptic terminals. The frequency of the mSCTs and their sensitivity to AP5 and alterations in spontaneous release supports their synaptic origin.

*7) Introducing Fluo-4 salt with a patch pipette reduced observation frequency of mSCTs and the authors interpret this as being due to wash-out. However, it seems that the pipette solution contained high gluconate, which is known to buffer Ca*^*2+*^
*(*K_*d*_
*∼35 mM). Although* K_*d*_
*may appear high it would be expected to affect [Ca*^*2+*^*] due to its very high concentration. Washout should produce a decrease in frequency over time compared to Fluo-4 AM, was this observed?*

Author response image 1.**DOI:**
http://dx.doi.org/10.7554/eLife.09262.014

We were concerned about wash out as well as photobleaching and therefore we limited our acquisition time to 2 min. This duration was also necessitated by the relatively high temporal resolution of our optical recordings.

Figure 7 shows that over the course of 4 experiments with Fluo-4 pentapotassium salt, we only detect a minor trend towards a decrease in frequency over the recording window.

8) Frequency of mSCTs measured with GCaMP5k-PSD95 was ∼1 very 2 hours. How long were the recordings?

These recordings were ∼2 minutes long due to their relatively high temporal resolution (10 frames per second) as well as due to the concerns indicated above.

*9) In the fourth paragraph of the subsection “Visualization of miniature spontaneous CA*^*2+*^
*transients in hippocampal neurons” and in*
Figure 1*: The authors claim that differences were insignificant except comparing GCaMP5k-PSD to Fluo-4 AM. From the error bars in*
Figure 1
*this seems not to be supported by the data.*

We agree with the reviewer that at a glance these data should be statistically significant. However, when we use the appropriate Holm-Sidak multiple comparison test, we only detect a significant difference between GCaMP5k-PSD95 and the Fluo-4 AM conditions. If we use pairwise *t*-tests, indeed these data all seem significant but we do not think this is appropriate way to compare. All experiments had relatively high numbers except for the impermeable Fluo4 injection out of 15 experiments only 4 yielded interpretable findings.

10) Does the lack of events with GCaMP5k-PSD indicate that these are extrasynaptic? If so, this should be stated more clearly.

GCaMP5K-PSD probe can detect synaptic Ca^2+^ transients in the absence of Mg^2+^ ([25]
Figure 2 therein). While the lower detection efficiency of GCaMP5K-PSD95 compared to soluble probes does not prove the location of these transients in itself, it does suggest that the bulk of the Ca^2+^ transients occur in a cellular compartment where it cannot be measured by the postsynaptically localized probe. We have added clarification in the last paragraph of the subsection “Visualization of miniature spontaneous Ca^2+^ transients in hippocampal neurons”.

In addition, the majority of Ca^2+^ transients measured with Fluo-4 AM are sensitive to ryanodine block which indicates that the ER is the bulk source of the Ca^2+^ ions for these events.

*11)*
Figure 4*: For these negative results – how did the authors ensure that the blockers actually worked in these experiments effectively blocking the respective receptors and L-type Ca current?*

NBQX is a common laboratory drug and is used concurrently in many studies in our lab. Its efficacy has never been problematic (e.g. [14], PNAS). Nimodipine, YM202074 and Fenobam were obtained for this project alone so we tested the efficacy of these compounds as follows: Nimodipine, the L-Type Ca^2+^ blocker was tested in our dissociated culture system by loading cells with Fluo-4 AM as elsewhere in the manuscript and in Tyrode’s solution containing 50 µM AP-5 and 20 µM NBQX. In the presence of sustained stimulation to evoke 10 action potentials at 0.2 Hz, the extracellular solution was changed with one that also contained 10 µM Nimodipine and 10 more action potentials were evoked. Our results show that there is a significant decrease in action potential dependent Ca^2+^ influx after the addition of Nimodipine.

To test whether the mGluR1/5 blockers YM202074 and Fenobam were functioning properly, we employed a similar setup using dissociated cultures loaded with Fluo-4 AM. Transient Ca^2+^ increases were evoked by perfusing the cells with 100 µM of the mGluR1/5 agonist (S) -3,5-Dihydroxyphenylglycine (DHPG). To test the efficacy of our blockers, baseline fluorescence was collected in solution containing YM202074 and Fenobam and then DHPG was added. During this experiment YM202074 and Fenobam were able to completely block the transient Ca^2+^ rise associated with DHPG (see Figure 4—figure supplement 1 and Figure 4—figure supplement 2).

*12) The legend of*
Figure 4*:* “*The remaining mSCTs in the KO animals are unaffected by AP-5*”*. This seems not to refer to the data shown in*
Figure 4*.*

“The remaining mSCTs in the KO animals are unaffected by AP5” is in the body of the Figure 3 legend. This sentence is in reference to the fact that the addition of AP- 5 does not decrease the mSCT frequency in the SNAP25 KO animals to zero but rather to very low levels. The remaining mSCTs, though few, persist even in the presence of AP-5.

*13)*
Figure 5
*shows a much larger effect of AP5 than ryanodine. How do the authors interpret this result?*

We have added the following to the text (subsection “Blocking mSCTs induces homeostatic eEF2 kinase-dependent synaptic scaling):

“This increase in mEPSC amplitudes was not as pronounced as was found with the positive control (TTX + AP5) which may correlate with the finding that ryanodine treatment (presumably due to its use-dependent nature) does not block mSCTs as completely as AP5 (Figures 2 and 4).”

Reviewer #2:

This is an interesting and timely paper on a topic that has received much recent attention in neuroscience. I believe that this paper should be accepted, but there is one somewhat large issue regarding sample size that need be addressed first:

*Throughout the paper, the sample size for the experiments is low (N of 5 to 6 is not uncommon). For example, in*
Figure 4*, which encompasses five separate experiments, the total N for all five experiments appears to be less than 30.*

*(This general issue, RE sample size and statistics in neuroscience, is discussed here:*
*http://www.nature.com/nrn/journal/v14/n5/full/nrn3475.html**). It seems that the N should be increased in some experiments reported here.*

We thank the reviewer for pointing these issues out. We now followed the reviewer’s advice and increased the N for the experiments presented in Figures 2, 3 and 4 (with the lowest N number of 7 for imaging experiments).

It is also worthwhile to note that from Figure 2 on, all of the imaging experiments are done in a before/after format which greatly decreases false findings and confounds by allowing us to contain comparisons within the same cell and use a paired *t*-test.

*In the first paragraph of the subsection “Visualization of miniature spontaneous CA*^*2+*^
*transients in hippocampal neurons”: The frequency of spontaneous Ca*^*2+*^
*transients was 0.32 per minute. If you sum the Ca*^*2+*^
*transients across all release sites in entire cell, do you get a Ca*^*2+*^
*transient frequency that roughly matches the mEPSC frequency (∼1Hz)? If so, that would be a good validation of the method. If not, then either there is a problem with the method or there's some fraction of release that does not trigger NMDAR/CICR signaling, which would be good to know. If this is not technically feasible, please explain.*

Spontaneous release rate per synapse has been estimated to be in the order of 1 vesicle per 60-90 seconds (Geppert et al., 1994 Cell; Murthy and Stevens, 1999 Nature Neuroscience, Sara et al., 2005 Neuron). Our recent best measurement of per synapse spontaneous release frequencies comes from a recent publication titled “Fast retrieval and autonomous regulation of single spontaneously recycling synaptic vesicles” ([25]
*eLife*) where presynaptic vGlut-pHluorin imaging yielded an average frequency of 0.76 ± 0.03 release events/synapse/minute. The relatively higher rate per synapse does in this case suggest that not every mEPSC is able to generate an mSCT. This rate comparison has been added to the Discussion.

Figure 5*: It is surprising that ryanodine had no effect on mEPSC frequency. From the representative traces, it would appear that ryanodine did have an effect. Maybe these are not very representative? Or, do the authors think there was a slight effect of ryanodine in mEPSC frequency that did not reach significance? Please address.*

The acute treatment of neurons with ryanodine did not alter mEPSC frequency. We have included this data to help illustrate that the primary effect of ryanodine in driving mEPSC scaling is the postsynaptic block of mSCT activity rather than driving a change in release frequency. This point has now been clarified in the Results section discussing Figure 5 (subsection “Blocking mSCTs induces homeostatic eEF2 kinase-dependent synaptic scaling”, second paragraph). Upon close inspection, the example trace shown in Figure 6 was not a good representative and has been replaced.

Reviewer #3:

*[…] The manuscript has several interesting findings, however one major issue is that there is quite a jump between the story on what is the source of spontaneous dendritic calcium transients to how it affects the synapse. Either they need to provide better transition or posit the paper in a different manner.*
Figure 1
*through 4 make sense together, but the last two figures, which are addressing synaptic scaling, appear without much of a transition.*

We thank the reviewer for this comment. We now revised the text to include a better transition and rationale for the homeostatic synaptic scaling experiments (subsection “Blocking mSCTs induces homeostatic eEF2 kinase-dependent synaptic scaling”, first paragraph).

*1) In*
Figure 1*, the calcium transient from Fluo-4 AM is not increased in 0 magnesium, the authors do point that out. In light of what is indicated in the Introduction, where they state '…that calcium signal is 20% of the full calcium signal of unblocked NMDA…', it is quite surprising that the signal is not higher in 0 magnesium. This needs to be addressed.*

It is indeed true that the addition of physiological concentrations of Mg^2+^ to the extracellular solution decreases the Ca^2+^ flux through the NMDA receptor significantly. The fact that removing Mg^2+^ from the extracellular solution did not yield larger mSCT amplitudes as measured by Fluo-4 AM was the first clue that these events were not solely dependent on NMDA activity. We have clarified the end of the first paragraph under the Results section to reflect this.

In order to more directly address the contribution of the NMDA receptor current to the observed Ca^2+^ transients we have added a new experiment as Figure 4.

We incubated neurons with ryanodine for 15 minutes to block RyRs and then loaded them with Fluo-4 AM as before. These cells were imaged in Tyrode’s solution containing TTX but no extracellular Mg^2+^ to allow maximal NMDA currents. Under these conditions Ca^2+^ transients were observed, but when 1.25 mM Mg^2+^ was again added no further transients could be measured. These results illustrate that under physiological concentrations of Mg^2+^, Fluo-4 cannot detect the NMDA Ca^2+^ transient without further amplification from ER Ca^2+^ stores.

Furthermore, because Fluo-4 AM detects calcium events both in the presynapse and postsynapse, an explanation or experiment is required to distinguish these two sources. How did they define the calcium events as dendritic only?

In Figure 2 the addition of AP5 was able to abolish the vast majority of mSCTs as detected by Fluo-4 AM. This finding is evidence that these mSCTs are postsynaptic as the distribution of NMDA receptors is highly postsynaptic.

We used multiple probes or delivery methods to detect these mSCTs (postsynaptic dialysis, sparse tranfection of GCAMP5 probes etc) to verify the postsynaptic origin of the vast majority of the events. They are NMDA receptor block sensitive. Our earlier experiments in the same system provided little to no evidence for a role for presynaptic NMDA receptors during spontaneous neurotransmitter release (Atasoy et al., 2008 Supplementary Figure 2).

Figure 1*, is the impermeable Fluo-4 experiment conducted in Syb2-mO infected neurons? If so, it is worth checking if there is any spontaneous transients in the presynaptic terminal of cells that express Syb2-mO.*

In our experiments to date we did not detect any pulsatile presynaptic Ca^2+^ release signals. In an earlier study, where we focused on presynaptic Ca^2+^ transients in presynaptic terminals (using a similar setting as in here), we found no evidence of discrete events but only of changes in baseline Ca^2+^ levels in response to neuromodulators such as Reelin (Refer to Bal et al., 2013 Neuron, Figure 3).

*2) The authors showed mSCT frequency is nicely correlated with mEPSC frequency, when influenced by presynaptic players (*Figure 3*). On the other hand intracellular calcium store blockers, Dantrolene and Ryanodine potentially only affecting the dendrite each reduce mSCT frequency (*Figure 4*). It is surprising and also hard to understand how calcium release from internal stores affects frequency rather than amplitude or kinetics of the calcium transients. It would be important to address if the presynapse is affected by Dantrolene and Ryanodine.*

Calcium release from internal stores generates virtually all of the calcium signal visible with Fluo-4 imaging. The reduction in mSCT frequency with ER store blockers represents the fact that mSCTs can no longer be generated in their presence. In order to clarify this point, we have changed the wording in the Results section for Figure 4 (subsection “mSCT generation requires Ca^2+^ release from internal stores”). The remaining mSCTs after treatment with ryanodine are in fact of reduced amplitude suggesting that the remaining mSCTs are partially blocked by this treatment.

With 1.25 mM Mg ^2+^ in the extracellular solution the signal detected by Fluo-4 AM is exclusively produced by ER store calcium. To illustrate this point we conducted an experiment where cells were pre-treated with ryanodine to block calcium release from ER stores. Cells were then imaged with and without magnesium to show both the maximal current flow for the NMDA receptor (no magnesium) and the physiological condition (1.25 mM magnesium). With no magnesium in the bath and the NMDA receptor current uninhibited, we were able to measure calcium transients after the treatment with ryanodine. However when magnesium was again washed over the cells no further transients could be detected (Figure 4). From these results we conclude that the diminutive NMDA current present in 1.25 mM magnesium is not detectable by our method.

*3)*
Figure 5*, the authors indicate that ryanodine and AP5 each only influence mEPSC amplitudes not the frequency. From the sample trace it appears that frequency is actually increased in both cases, a more representative trace is needed here.*

As we indicated in our response to reviewer #2’s comments, the acute treatment of neurons with ryanodine did not alter mEPSC frequency. We have included this data to help illustrate that the primary effect of ryanodine in driving mEPSC scaling is the postsynaptic block of mSCT activity rather than driving a change in release frequency. This point has now been clarified in the Results section discussing Figure 5 (subsection “Blocking mSCTs induces homeostatic eEF2 kinase-dependent synaptic scaling”, second paragraph). Upon close inspection, the example trace shown in Figure 6 was not a good representative and has been replaced.

4) Why does ryanodine affect the calcium transient frequency, but affects only the mEPSC amplitude? This seems contradictory. I would expect an effect on the mEPSC frequency as well.

From our results, we believe that ryanodine acts downstream of mEPSC activity but its impact on generated Ca^2+^ transients and local protein translation alters mEPSC amplitudes. In other words, mSCTs largely report CICR driven Ca^2+^ signals that are triggered but not tightly coupled to the mEPSCs (which mainly report AMPA receptor driven currents). This issue is now better stated in the text where we discuss the impact of acute ryanodine treatment on mEPSC frequency (subsection “Blocking mSCTs induces homeostatic eEF2 kinase-dependent synaptic scaling”, second paragraph).